# Relationship between Lifestyle and Residence Area with 25(OH)D Levels in Older Adults

**DOI:** 10.3390/ijerph20010407

**Published:** 2022-12-27

**Authors:** Ubiraídys de Andrade Isidorio, Elisangela Vilar de Assis, Sheylla Nadjane Batista Lacerda, Ankilma do Nascimento Andrade Feitosa, Beatriz da Costa Aguiar Alves, Thais Gascón, Glaucia Luciano da Veiga, Fernando Luiz Affonso Fonseca

**Affiliations:** 1Departamento de Ciências da Saúde, Faculdade Santa Maria de Cajazeiras, Cajazeiras 58900-000, Brazil; 2Laboratório de Análises Clínicas, Centro Universitário FMABC, Santo André 09060-870, Brazil; 3Departamento de Ciências Farmacêuticas, Universidade Federal de São Paulo, Diadema 09972-270, Brazil

**Keywords:** healthy lifestyles, prevention health risks, 25(OH)D, elderly

## Abstract

The aim of this study was to investigate whether the habitat of older adults living in the urban and rural areas is a contributing factor to altered serum 25(OH)D levels. An analytical, cross-sectional research with a quantitative approach was conducted in the upper backwoods of Paraíba with older adults registered at the Reference Center for Social Assistance (CRAS), addressing residents of both rural area (*n* = 41) and urban area (*n* = 43) who were randomly chosen to participate in the study. Data collection took place between January and February 2020. Higher rates of handgrip strength were observed in rural older adults (29.22 ± 8.92 Kgf) with serum 25(OH)D levels in the range of 30 to 40 ng/mL when compared to older adults with concentrations below 30 ng/mL (21.82 ± 5.00 Kgf) and above 40 ng/mL (23.47 ± 6.88 Kgf). Older people living in the urban area, with 25(OH)D levels above 40 ng/mL, presented a lower muscle power index when compared to individuals with vitamin levels from 21 to 29 ng/mL (17.40 ± 6.34 s; 15.33 ± 2.00 s). The urbanization associated with 25(OH)D levels can cause changes in skeletal and respiratory muscle strength, because the habitat associated with 25(OH)D levels affects the parameters of muscle strength of the upper limbs for older adults living in a rural area and the muscle strength of the lower limbs for those living in an urban area.

## 1. Introduction

The world’s elderly population has been growing significantly in recent years and is associated with aging and organic functional decrease, which occur proportionally over time [1]. In addition to the physiological changes relevant to this process, older adults may also adopt undue health practices, such as sedentary lifestyle and bad eating habits, favoring the triggering of diseases and functional limitations [2].

Reduced mobility is one of the main causes related to musculoskeletal dysfunctions in which there is a type II muscle fiber atrophy, which contributes to impaired muscle coordination and functional disability, resulting in increased morbidity and mortality. This fact makes older people more vulnerable and restricts their activities in the social environment, often compromising their quality of life [2,3].

Muscle power has a direct relationship with functional exercise capacity and gait speed, which are factors inherent to an adequate ambulation that decreases the prevalence of falls resulting in fractures with significant impact on hospital costs and health quality of these individuals [3].

The decreased muscle strength occurs similarly in the respiratory musculature. Thus, age is a negative indicator of respiratory muscle forces for both men and women [4]. It can generate a reduction in rib cage mobility, pulmonary elasticity, and respiratory muscle strength, which happens gradually and significantly to each older person [5].

One of the factors observed among older adults that can contribute to the reduction of muscle strength is the decrease in 25(OH)D. In the muscle, this hormone favors increased strength, as well as contributes to postural and dynamic balance, and is associated with increased bone mineral mass and subsequent prevention of fractures arising from fall [6].

25(OH)D insufficiency among older adults is a common clinical condition and frequently occurs due to eating habits, intestinal absorption, black skin, air pollution, smoking, and reduced ability to synthesize this vitamin by the skin [7,8]. In addition to these factors, the endogenous synthesis of 25(OH)D is dependent on sun exposure, and it is related to latitude [8].

Thus, there arises the importance of investigating whether the habitat of older adults living in the urban and rural areas is a contributing factor to altered serum 25(OH)D levels, highly affecting the skeletal muscle strength of these individuals.

## 2. Materials and Methods

An analytical, cross-sectional research with a quantitative approach was conducted in the upper backwoods of Paraíba with older adults who were registered at the Reference Center for Social Assistance (CRAS). Rural (*n* = 41) and urban (*n* = 43) residents were randomly chosen to participation. Data collection took place between January and February 2020.

The participants were older adults of both sexes, who were aged over 60 years and consented to participation. This study excluded older adults with cognitive impairment and visual, cardiac, respiratory, renal, and neurological disorders, in addition to those with neoplasm and those who were bedridden.

This study was approved by the Research Ethics Committee at the College of Santa Maria under protocol number. 3,376,161, complying with Resolution 466/12 of the National Health Council regarding research involving human beings [9]. Personal data, sociodemographic data, personal history (respiratory, heart, and renal disorders; arterial hypertension; and diabetes mellitus), life habits, and general health status were collected. Anthropometric measurements, skeletal muscle strength (handgrip strength, muscle power, and respiratory muscle strength), and laboratory tests were also obtained to quantify 25(OH)D levels.

The Body Mass Index (BMI) was obtained and, in accordance with the values established by the World Health Organization (WHO), was calculated by dividing mass (Kg) by height squared (m^2^) [10,11]. Abdominal circumference (AC), hip circumference (HC), and waist–hip ratio (WHR) were obtained, with the latter being classified according to the WHO and calculated by dividing the AC by the HC, which values below 0.85 are indicative for women and 1.0 for men; a higher value expresses a higher risk for cardiovascular diseases [12,13].

Respiratory muscle strength (RMS) data, maximal inspiratory pressure (MIP), and maximal expiratory pressure (MEP), adjusted in cmH_2_O and with a variation of ±300 cmH_2_O, were obtained using the Wika^®^ analog manovacuometer device (São Paulo, Brazil); the participants were evaluated when seated, with the spine upright and the trunk at 90° in relation to the hip [14,15]. Three measurements were performed, and the highest value was adopted if the difference between the values was not above 10% [5].

During the maneuvers, the participants received standardized verbal stimuli. The following intervals were adopted: 15 s between the measurements; 1 min between the tracheal change; and 30 s between the maneuvers [16].

Handgrip strength was assessed using the Manual Camry^®^ (San Diego, CA, USA) Digital Dynamometer, with the participants seated, their elbow flexed at 90°, and their forearm and wrist in a neutral position, while performing three maximal isometric contractions, with a brief pause between the measurements. Three measurements of each hand (dominant and non-dominant) were obtained, and the arithmetic mean was calculated, grouping the measurements to indicate the dominant and non-dominant hand [17].

Lower-limb muscle power was assessed by performing the sit-and-stand test 5 times, in which the participants stayed seated, arms crossed on the chest and leaning against the back of a chair. The following instructions were provided: lift and sit five times as fast as you can when I say, “go”. The time was measured from the “go” sign until the end of the five repetitions using a digital stopwatch by Lab-Commerce. To analyze lower-limb strength/power, the cutoff points described by Bohannon et al., (2012) were used: 11.4 s for people aged 60 to 69 years old; 12.6 s for people aged 70 to 79 years old; and 12.7 s for people aged 80 to 89 years old [18].

Venous blood (10 mL in vacutainer tubes with and without anticoagulant) was collected from the participants for laboratory determination of 25-hydroxyvitamin D concentration.

The samples were centrifuged at 2500 RPM for 10 min at 4 °C. Serum and plasma were separated and stored in a refrigerator until laboratory tests were performed. 

The concentration of 25(OH)D was determined using the electrochemiluminescence method in an immunoenzymatic assay, using a standardized reagent in a COBAS 8000 equipment (Roche^®^, Basel, Switzerland). The clinical analyses were performed in accordance with good practices at all stages of the laboratory evaluation. 

25(OH)D was evaluated using the blood samples collected on a pre-scheduled day and time, which were sent to a private laboratory for analysis, using the cutoff points established by the American guideline for evaluation, prevention, and treatment of 25(OH)D deficiency: deficiency is below or equal to 20 ng/mL; insufficiency is 21–29 ng/mL; and sufficiency means equal to or above 30 ng/mL [19].

The results are shown in graphs and tables. Categorical variables are described in absolute (*n*) and relative (%) frequencies and analyzed using the statistical software EPI INFO^®^ version 7.2.3.1, using the Chi-square test. Quantitative variables are expressed as mean and standard deviation. These analyses were performed using the statistical software BioEstat^®^ version 5.0 (Mamirauá Institute, Belém, Brazil). The Shapiro–Wilk normality test was used, followed by the Mann–Whitney Test or Variance Analysis (ANOVA) and Tukey’s post hoc test, considering differences to be significant when *p* < 0.05.

## 3. Results

The mean age was 69 years for the participants from the rural area and 70 years for those from the urban area. The majority of participants for both areas are of pardo ethnicity (66% and 56%), have incomplete elementary education (51% and 58%), have per capita income up to one minimum wage (98% and 95%), and are retired (100% and 93%) (Table 1).

Table 2 shows the characterization of the participants in the rural and urban areas according to the presence of pathologies, behavioral habits, hospitalization, falls, and perceived health. Male individuals prevailed in the rural area (39%) when compared to the urban area (16%) (*p* < 0.05). The rural older adults were composed of a greater number of married individuals (66%), whereas those residing in the urban area were mostly widowed (56%), (*p* < 0.05) (Table 1).

There were no differences between the rural and urban older people regarding the presence of diabetes mellitus, hypertension, physical activity, and alcohol use (*p* > 0.05) (Table 2).

Hypertension was the most reported health problem, affecting 56% of the rural older adults and 56% of the urban older adults. Only 27% of the rural older adults and 23% of the urban older adults practiced physical exercises twice or more a week. More than 90% of the rural and urban older adults did not consume alcoholic beverages. The rural older adults had a lower percentage of smokers (7%) and a higher percentage (80%) of good self-perceived health (*p* < 0.05), when compared to the urban individuals (Table 2).

Table 3 illustrates the distribution of the participants according to anthropometric measurements and muscle and respiratory strength parameters, according to the area of residence and sex.

BMI, maximal inspiratory and expiratory pressure, and 25(OH)D levels did not differ between the two groups analyzed (*p* > 0.05). The waist/hip ratio for both sexes was higher in the rural older adults (*p* < 0.05). There was no difference between the rural and urban older adults concerning handgrip strength, but the strength intensity was higher in males (32 Kgf and 29 Kgf) when compared to females (22 Kgf and 20 Kgf) (*p* < 0.05). In the sit-and-stand test, rural women presented higher muscle power levels (18 s) when compared to urban women (14 s), unlike males (*p* < 0.05) (Table 3).

The older adults are also distributed according to aspects of muscle and respiratory strength and 25(OH)D levels (Figure 1). There were significant differences in handgrip strength between the rural (29 ± 9 Kgf) and urban (22 ± 6 Kgf) older adults with 25(OH)D concentrations from 30 to 40 ng/mL (*p* < 0.05). Higher rates of handgrip strength were observed in the rural older adults (29 ± 9 Kgf) with serum 25(OH)D levels ranging from 30 to 40 ng/mL when compared to the older adults with concentrations below 30 ng/mL (22 ± 5 Kgf) and above 40 ng/mL (23 ± 7 Kgf) (*p* < 0.05) (Figure 1).

The relationship between muscle strength and 25(OH)D levels in the rural and urban older adults was analyzed. There were differences between muscle power for the groups with serum 25(OH)D levels of 30 to 40 ng/mL (17 ± 5; 14 ± 2 s) and with levels above 40 ng/mL (18 ± 3 s *; 13 ± 1) (* *p* < 0.05). The urban older adults, with 25(OH)D levels above 40 ng/mL, presented a lower muscle power index when compared to individuals with vitamin levels from 21 to 29 ng/mL (17 ± 6; 15 ± 2 s *).

Figure 2 describes the MIP and MEP levels. The MIP levels for the rural residents were higher for individuals with serum 25(OH)D concentrations of 30 to 40 ng/mL (88 ± 15 cm H_2_O) and > 40 ng/mL (89 ± 10 cm H_2_O), when compared to those in the group with 21 to 29 ng/mL (74 ± 9 cm H_2_O) (Figure 3A). There were no significant differences in the MEP scores between the groups analyzed (Figure 3B), *p* > 0.05.

Figure 3 shows the Pearson’s correlation analysis performed between HGS, PImax, Pemax, and 25(OH)D levels.

## 4. Discussion

Brazil is facing an accelerated aging population that has emerged in a context marked by few resources and social inequalities. The Brazilian elderly population consists mostly of females (54%), with less than eight years of schooling (85%), who are affected by chronic diseases, such as hypertension (63%) and diabetes (16%), as shown in the results described in this study. Moreover, around 24.1% of older adults live in northeastern Brazil and 15% live in rural areas [20].

The urbanization of older adults and an aging population bring new challenges, adding to the need to develop strategies that can ensure that this phase of life is lived preferably with independence, autonomy, and quality of life, thus affecting the health of older adults [21].

A fundamental aspect for promoting the health of older adults is a healthy lifestyle, which promotes the maintenance of the individual’s functional capacity and life satisfaction. Therefore, the place of residence is one of the factors that determine the lifestyle of older adults. It is noteworthy that, regardless of the region of residence (urban or rural), most of the older adults may have a good lifestyle, with the maintenance of healthy habits and prevention of functional limitations [22].

A study with older people in southern Brazil found similar percentages of elderly smokers living in the rural (13%) and urban areas (14%) [23]. On the other hand, a Brazilian study, which evaluated the importance of the urban-rural context in determining the pattern of consumption of tobacco products, illustrated that, despite the knowledge that the degree of urbanization was associated with the consumption of tobacco products, the places least influenced by the urban environment had a higher prevalence of smoking and smoking cessation was higher in the more urbanized places [24].

Self-perceived health is an essential parameter that influences the quality of life of older adults, since when people define their own health as poor, the majority (91%) also presents poor quality of life [25]. A cross-sectional study conducted in the state of Paraná, Brazil, highlighted that older adults in the rural area, when compared to those in the urban area, reported better overall quality of life scores, which mainly encompassed physical, psychological, social, relationship, autonomy, activity and social participation domains [21].

The waist/hip ratio is also a measure used to assess abdominal obesity, which can cause vulnerabilities to physical health, and its scores may vary according to age, gender, and habitat. In China, waist-to-hip ratio scores have been found to be higher in women in rural areas and their values increase with age [26]. A study that evaluated anthropometric measures and body composition reported similar results, i.e., with no significant differences between the BMI of residents of rural and urban areas, but the waist/hip ratio was higher in residents of rural areas [27].

Vitamin D supplementation promotes the reduction of anthropometric parameters, such as BMI and waist circumference [28]. However, the mechanisms of action of vitamin D involved in the pathogenesis of obesity are not yet clearly defined, thus making studies in this field of knowledge necessary [29].

A Brazilian study on hypovitaminosis in women and newborns described that insufficient levels of 25(OH)D were not associated with the place of residence (rural or urban) [30]. Nevertheless, different results have been reported in international studies.

Although rural older adults spend more hours outdoors than older adults in urban areas, vitamin D levels between urban and rural regions may be similar if both have levels of sun exposure in their daily routine long enough to maintain the desired levels of vitamin D [31]. Since excess sunlight does not linearly increase vitamin D production, sun exposure greater than 1 h daily can keep serum vitamin D levels above 50 nmol/L [32].

Solar incidence induces the synthesis of vitamin D, which is mainly influenced by the latitude and altitude of a region, season, time of sun exposure, and aging. During exposure to sunlight, 7-dehydrocholesterol in the skin absorbs UV B radiation, which is converted into pre-vitamin D3, giving rise to photoproducts, such as 25(OH)D3, which will be metabolized in the liver and kidneys, synthesizing vitamin D [33].

The results of this study corroborate with the scientific literature. There are reports that vitamin D3 positively impacts muscle strength, and strength measures significantly improve after supplementation [34,35,36,37]. Moreover, in older adults, vitamin D supplementation has shown a reduction in the risk of falls and improvements in muscle performance tests, since vitamin D and its receptors are fundamental for the normal development of skeletal muscle and for the optimization of muscle strength and performance [38].

Analyzing specifically HGS, Wang et al. (2019) in a cohort study evaluated men over 50 years old with accelerated and progressive loss of HGS and, interestingly, found a positive correlation between 25(OH)D concentrations and HGS in these individuals, which was not found in men below this age. These data confirm 25(OH)D’s direct participation in the improvement of HGS in older adults and justify the supplementation of this vitamin in this age bracket, in addition to the benefits of bone improvement [35]. Other studies have also verified this association between 25(OH)D levels and increased HGS, specifically for older individuals. This increase in HGS is independent of the practice of physical activity [36]. It is known that HGS is an important tool to identify mobility limitations in clinical practice [37], which makes the change in this variable even more significant in older individuals with higher levels of 25(OH)D.

Similar results regarding handgrip strength indices were reported in a study with rural older people in the state region of Bahia, with females having a lower handgrip strength (17 kg) when compared to males (25 kg) [39]. In a study with older people assisted by primary health care services in the state of Paraná, the mean handgrip strength of these older adults was 29 Kgf, and males had higher values (39 Kgf) when compared to females (24 Kgf) [40].

The differences between genders found in this study are probably associated with the fact that men have a higher muscle mass reserve than women. In men, the differences in the percentage of skeletal muscles are frequently more concentrated in the upper limbs than in the lower limbs [41].

Muscle strength parameters are important in clinical practice to identify subgroups of frail older people and identify loss of muscle strength, including loss of respiratory muscle strength, which can be assessed through maximal inspiratory pressure (MIP) and maximal expiratory pressure (MEP) [42].

There are still gaps in the scientific literature regarding the interference of the association of 25(OH)D and habitat in respiratory muscle strength parameters in older adults. However, for decades, the knowledge that 25(OH)D plays a role far beyond bone health has been widespread; it contributes to the respiratory system by mechanisms involved in inhibiting pro-inflammatory pulmonary responses and improving the defense mechanisms of the innate immune system against respiratory pathogens [43].

Currently, the action of 25(OH)D and respiratory muscle strength parameters have been the target of studies within the context of the global pandemic resulting from coronavirus infection.

Infection caused by COVID-19 may cause impaired performance of respiratory muscles, but respiratory muscle performance measures are not routinely performed in clinical practice [44].

In this context, together with clinical parameters indicative of respiratory muscle strength, 25(OH)D levels should also be investigated since 25(OH)D has been pointed out as a potential agent to be used in the fight against SARS-CoV-2. It has the ability to reduce respiratory tract infections through the induction of catelicidines and defensins that can reduce the rates of viral replication and pro-inflammatory cytokines, easing the picture of pulmonary inflammation [45].

## 5. Conclusions

In conclusion, the urbanization associated with 25(OH)D levels can cause changes in skeletal and respiratory muscle strength, because the habitat associated with 25(OH)D levels can affect the parameters of muscle strength of the upper limbs among older adults in the rural area and muscle strength of the lower limbs among those in the urban area.

The older adults from the rural area with insufficient serum levels of 25(OH)D presented reduced handgrip strength, whereas, for urban residents, the high levels of 25(OH)D correlated with lower muscle power scores of the lower limbs when compared to individuals with sufficient 25(OH)D levels.

The maximal inspiratory pressure scores were reduced among the older adults in the rural area, who had insufficient levels of 25(OH)D, when compared to the other groups. Thus, there is a need to monitor serum 25(OH)D levels in older adults to avoid insufficiency or excess of supplementation.

## Figures and Tables

**Figure 1 ijerph-20-00407-f001:**
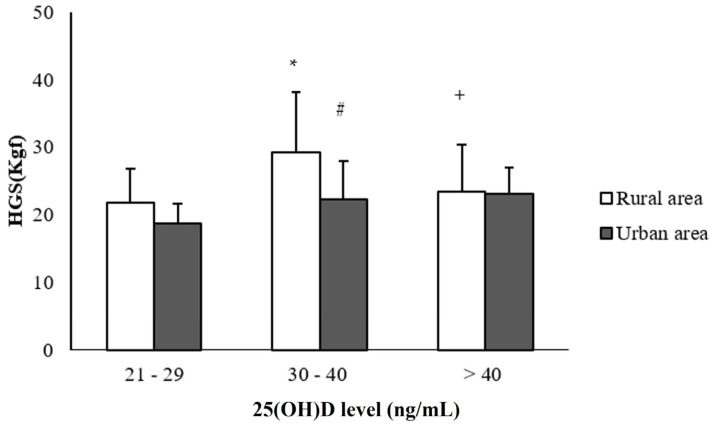
Distribution of older adults in the rural and urban areas according to handgrip strength and 25(OH)D levels. # Statistical difference between the groups of older adults in the rural versus urban areas, considering the same level of 25(OH)D, *p* < 0.05. * Statistical difference in relation to the group with levels from 21 to 29 ng/mL, considering the same area of residence, *p* < 0.05. + Significant differences, *p* < 0.05, in relation to the 30 to 40 ng/mL group, considering the same area of residence. The differences were analyzed using the ANOVA test, followed by the Tukey’s test. The group with 21 to 29 ng/mL: *n* = 10 rural area and *n* = 9 urban area; the group of 30 to 40 ng/mL: *n* = 22 rural area and *n* = 26 urban area; and the group > 40 ng/mL: *n* = 9 rural area and *n* = 8 urban area. # *p* < 0.001; * *p* < 0.02; + *p* < 0.02.

**Figure 2 ijerph-20-00407-f002:**
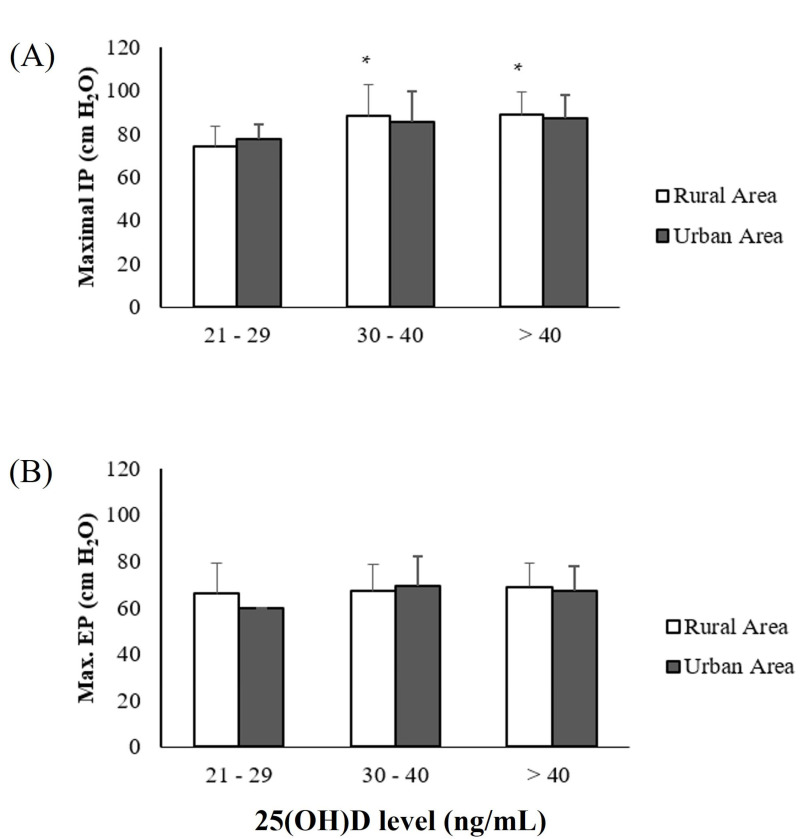
Distribution of older adults in the rural and urban areas according to maximal inspiratory and expiratory power pressures and levels of 25(OH)D. (**A**) Maximal inspiratory force (MIP) in cm H_2_O, and (**B**) maximal expiratory force (MEP) in cm H_2_O. * Statistical difference between the level of 25(OH)D in relation to the group with 21 to 29 ng/mL, considering the same area of residence, *p* < 0.05. The differences were analyzed using the Mann–Whitney variance test. A—MIP = * *p*= 0.02—rural area group: 30 to 40 × 21 to 29 ng/mL and *p* = 0.02 > 40 vs. 21 to 29 ng/mL.

**Figure 3 ijerph-20-00407-f003:**
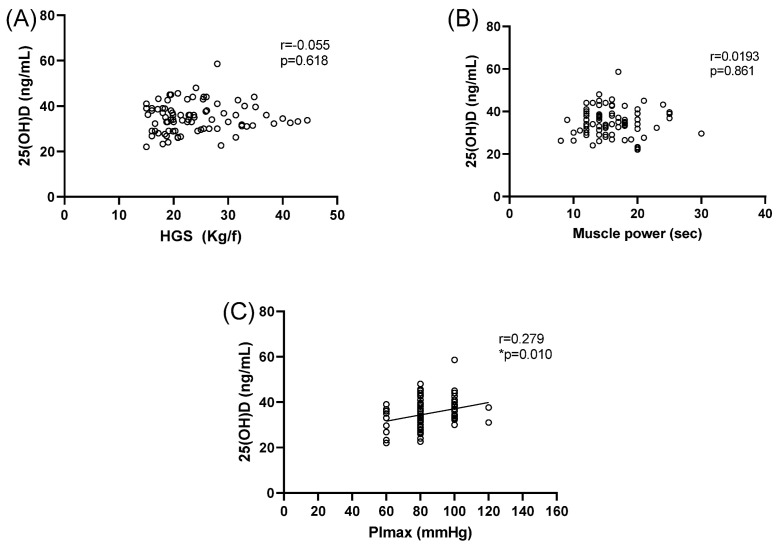
Pearson’s correlation between 25(OH)D concentrations and handgrip strength (HGS) (**A**), 25(OH)D concentrations and muscle power (**B**), and 25(OH)D concentrations and PImax (**C**). Pearson correlation test. * *p* < 0.05.

**Table 1 ijerph-20-00407-t001:** Distribution of the older people according to the sociodemographic characteristics.

Variables	Rural Area 41 (100%)	Urban Area 43 (100%)
Age (years) (Mean ± SD)	68 ± 6	70 ± 8
**Sex**		
Male	16 (39)	7 (16) ^#^
Female	25 (61)	36 (84)
**Ethnicity**		
White	12 (29)	14 (32)
Pardo	27 (66)	24 (56)
Black	2 (5)	4 (9.30)
Yellow	-	1 (2)
**Marital Status**		
Married/Stable union	27 (66)	8 (19)
Widow(er)	10 (24)	24 (56)
Separated	1 (2)	8 (19)
Unmarried	3 (7)	3 (7)
**Education**		
Illiterate	18 (44)	16 (37)
Incomplete elementary education	21 (51)	25 (58)
Complete elementary education	1 (2)	1 (2)
Incomplete secondary education	1 (2)	1 (2)
**Per capita income**		
Up to 1 minimum wage	40 (98)	41 (95)
Above one minimum wage	1 (2)	1 (2)
Blank	-	1 (2)
**Professional activity**		
Retired	41 (100)	40 (93)
Farmer	-	1 (2)
Hawker	-	1 (2)
Blank	-	1 (2)

^#^ Statistical difference verified by the Chi-square test, *p* < 0.05. Sex: *p*-value = 0.0019; Marital status: *p* = 0.0001.

**Table 2 ijerph-20-00407-t002:** Data about the comorbidities of the participants.

Variables	Rural Area 41 (100%)	Urban Area 43 (100%)
**Diabetes mellitus**		
Yes	5 (12)	7 (16)
No	36 (88)	36 (84)
**Systemic arterial hypertension**		
Yes	23 (56)	24 (56)
No	18 (44)	19 (44)
**Practice of physical activity**		
Yes	11 (29)	10 (23)
No	30 (73)	33 (77)
**Frequency of physical activity**		
Twice or more a week	11 (27)	10 (23)
**Alcohol use**		
Yes	1 (2)	3 (7)
No	40 (98)	40 (93)
**Smoking**		
Smoker	3 (7)	12 (28) ^#^
Ex-smoker	21 (51)	16 (37)
Never smoked	17 (41)	15 (35)
**Perceived health**		
Good	33 (80)	22 (51) ^#^
Regular	8 (19)	21 (49)

^#^ Statistically significant difference verified by the Chi-square test, *p* < 0.05. Smoking: *p* = 0.04; Perceived health: *p* < 0.001.

**Table 3 ijerph-20-00407-t003:** Characterization of the older adults according to anthropometric measures, muscle strength, and 25(OH)D levels according to place of residence and sex.

Variables	Rural Area		Urban Area	
	Male 16 (100%)	Female 25 (100%)	Male 7 (100%)	Female 36 (100%)
BMI (Kg/m^2^)	25 ± 4	26 ± 4	27 ± 5	25 ± 5
Waist/hip ratio	0.98 ± 0.05	0.94 ± 0.07	0.91 ± 0.05 ^#^	0.88 ± 0.08 ^#^
Vit D (ng/mL)	35 ± 8	34 ± 7	37 ± 5	35 ± 6
Handgrip strength (Kgf)	32 ± 7	22 ± 6 *	29 ± 5	20 ± 4 *
Muscle power (s)	16 ± 4	18 ± 5	13 ± 2	14 ± 2 ^#^
MIP (mmHg)	87 ± 14	83 ± 14	89 ± 16	83 ± 12
MEP (mmHg)	67 ± 13	66 ± 11	69 ± 15	67 ± 11

Mean (±SD) values are described. ^#^ Statistical difference between the groups of older people in rural × urban areas, considering the same sex, *p* < 0.05. * Statistical difference between male and female, considering the same area of residence, *p* < 0.05. The differences were analyzed using the ANOVA test, followed by the Tukey’s test; Waist/hip ratio = ^#^
*p* = 0.01 for males and *p* = 0.04 for females; Handgrip strength (Kgf) = * *p* < 0.001 for the rural area and *p* < 0.001 for the urban area; Muscle power = ^#^
*p* < 0.001 for females.

## Data Availability

All data from this study are available, if interested, please contact co-author Glaucia Luciano da Veiga (grlveiga@gmail.com).

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
