# Peer review of "Relationship between Lifestyle and Residence Area with 25(OH)D Levels in Older Adults"

_ijerph, 2022, doi:10.3390/ijerph20010407_

Round 1

Reviewer 1 Report

The manuscript entitled

Relationship between Lifestyle and Residence Area with Vitamin D Levels in Elderly by Glaucia Luciano da Veiga et al showed a nice presentation about vitamin levels of three aged mapped classes of elderly people living in urban or rural area. Slightly little differences vitamin D levels causes therefore different output in muscular and skeletal strength, e.g. hand grip rates.

But at least there are some addition needed in the manuscript which are missed.

Which type of methods was used for the detection of vitamin D. It is only written that this was measured in a private laboratory.

Also which type of Vitamin D was used; I suggest that this is Vitamin D3 (cholecalciferol) or calcidiol or –triol?

Which type of blood was collected (plasma, serum etc.).

Figures: Figure 2 the significances > 40 ng/mL Vitamin D are covering (#, *);

Also y-axis should involve more scale marks.

The difference of vitD levels differs not so much as expected between elderly people of urban or rural areas. Supplementation of vitamin D can be excluded? Which time of the year the study was done because endogenous production of vit D and does this involve both groups?

Infections can decrease vit D levels because of of its uptake in leucocytes:  CRP, leucocytes are similar in all groups?

Author Response

Response to Reviewer 1 Comments

Revisor 1#

Dear reviewer, we greatly appreciate your willingness to contribute to improving the understanding of our data.

Listed below are the responses, point by point, to the questions sent to us.

Additionally, we inserted the requests made into the article, and all of them were underlined in yellow.

Point 1: Which type of methods was used for the detection of vitamin D. It is only written that this was measured in a private laboratory.

 Response 1: The method was entered in the Materials and Methods section (page 3 line 105-108).

Point 2: Also which type of Vitamin D was used; I suggest that this is Vitamin D3 (cholecalciferol) or calcidiol or –triol?

Response 2: It was used Vitamin D3 (colecalciferol).

Point 3: Which type of blood was collected (plasma, serum etc.).

Response 3: It was collected Venous blood (10 mL in vacutainer tubes with and without anticoagulant) for determination of 25-hydroxyvitamin D concentration. (Line 101-104)

Point 4: Figures: Figure 2 the significances > 40 ng/mL Vitamin D are covering (#, *); Also y-axis should involve more scale marks.

Response 4: Figure 2 was removed and the data remained in the text for a better understanding of the data.

Point 5: The difference of vitD levels differs not so much as expected between elderly people of urban or rural areas. Supplementation of vitamin D can be excluded? Which time of the year the study was done because endogenous production of vit D and does this involve both groups?

Response 5: The data were obtained in the Brazilian summer.

Brazil is experiencing an epidemic of vitamin D hypovitaminosis, despite the constant incidence of sunlight. New studies have been carried out to evaluate the particularities of Brazilians regarding adequate vitamin D supplementation. Until the publication of new data about this profile, it is up to the physician to evaluate each case in particular and administer controlled supplementation.

Point 6: Infections can decrease vit D levels because of of its uptake in leucocytes:  CRP, leucocytes are similar in all groups?

Response 6: We agree with this placement, however, these data were not collected in the present study.

Reviewer 2 Report

Interesting study but needs more care in terms of how what was measured relates to vitamin D and other factors.

I suggest making scatter plots of:

HGS vs. 25(OH)D

Muscle power vs. 25(OH)D

MIP and MEP vs. 25(OH)D

With urban and rural values indicated by different symbols

Please state the locations of the two populations in terms of name of the city or rural region and latitude/longitude.

A discussion of factors that affect handgrip strength for the elderly should be given.

Also, what is the correlation between 25OHD and handgrip strength from other studies?

A search of scholar.google.com finds a few relevant articles

Vitamin D is related to handgrip strength in adult men aged 50 years and over: A population study from the TCLSIH cohort study

J Wang, X Wang, Y Gu, M Liu, VTQ Chi… - Clinical …, 2019 - Wiley Online Library

… Serum 25(OH)D concentration was significantly related to HGS in males aged above …
factors. Future studies are needed to clarify the age and sex relationship between serum 25(OH)D …

Save Cite Cited by 27 Related articles All 4 versions

Factors related with handgrip strength in elderly patients

N Riviati, S Setiati, PW LaksmiM Abdullah - Acta Med Indones, 2017 - staff.ui.ac.id

… The handgrip strength examination is often applied as a sarcopenia filtering … handgrip
strength. Methods: a cross-sectional study to determine factors related to the handgrip strength in …

Save Cite Cited by 64 Related articles All 11 versions 

Handgrip strength and associated sociodemographic and lifestyle factors: a systematic review of the adult population

TR de LimaDAS SilvaJAC de Castro… - Journal of bodywork and …, 2017 - Elsevier

… Through systematic gathering of scientific evidence about the association between handgrip
strength and sociodemographic and lifestyle factors, this research will serve as a support for …

Save Cite Cited by 47

Handgrip strength cutoff points to identify mobility limitation in community-dwelling older people and associated factors

KS de Souza Vasconcelos, JM Domingues Dias… - The journal of nutrition …, 2016 - Springer

… To determine the best cutoff point of handgrip strength for identifying mobility limitation
and to investigate the factors associated with muscle weakness and mobility limitation in …

Save Cite Cited by 72 Related articles All 9 versions

Factors associated with loss of handgrip strength in long-lived elderly

MH Lenardt, CRB Grden, JAV Sousa… - Revista da Escola de …, 2014 - SciELO Brasil

… Handgrip strength as a predictor of functional, psychological and social health. A … was to
investigate the prevalence of reduced handgrip strength and associated factors in the long-lived …

Save Cite Cited by 32 Related articles All 14 versions 

Handgrip strength and associated factors in hospitalized patients

RS Guerra, I Fonseca, F Pichel… - Journal of Parenteral …, 2015 - Wiley Online Library

… between hand dimensions and HGS. To our knowledge, this is the first study to show that
hand … Despite the adjustment of the dynamometer grip handle to the participant's handhand …

Save Cite Cited by 81 Related articles All 6 versions

Hand grip strength: outcome predictor and marker of nutritional status

K Norman, N Stobäus, MC Gonzalez, JD Schulzke… - Clinical nutrition, 2011 - Elsevier

… Among all muscle function tests, measurement of hand grip strength has gained attention as
… factors on hand grip strength. In acute or chronic disease, however, various further factors …

Save Cite Cited by 1054 Related articles All 11 versions

Sustained effect of resistance training on blood pressure and hand grip strength following a detraining period in elderly hypertensive women: a pilot study

DC Nascimento, RA TibanaFM Benik… - … interventions in aging, 2014 - Taylor & Francis

… most prevalent modifiable risk factor with a high prevalence among older adults. Exercise is
… prior to the measurement days. All measurements of BP and grip strength were performed …

Save Cite Cited by 62 Related articles All 21 versions

The solar incidence induces the synthesis of vitamin D, which is mainly influenced

252

by the latitude and altitude of the region, season, time of sun exposure and aging

Comment: These papers might be cited

The relationship between ultraviolet radiation exposure and vitamin D status.

Engelsen O.Nutrients. 2010 May;2(5):482-95. doi: 10.3390/nu2050482. Epub 2010 

Aging decreases the capacity of human skin to produce vitamin D3.

J MacLaughlin, MF Holick - The Journal of clinical …, 1985 - Am Soc Clin Investig

… To determine whether aging affected the capacity of human skin to synthesize vitamin D3,
we determined (a) the concentration of 7-dehydrocholesterol (provitamin D3) in a defined area …

Save Cite Cited by 1845 Related articles All 10 versions

giving rise to photoproducts such as vitamin D3, which

255

will be metabolized in the liver and kidneys, originating vitamin D [33].

Comment: That statement is not quite correct. One problem is that since “vitamin D level” is used instead of “25(OH)D concentration”, the meaning of vitamin D is ambiguous. The liver converts vitamin D3 to 25(OH)D; the kidney converts 25(OH)D to 1,25-(OH)2D3.

The differences between genders found in this study are probably associated with

269

the fact that men have a higher muscle mass reserve than women.

Comment: The role of testosterone should be mentioned and reference provided.

Regarding vitamin D and lung function:

A brief search at scholar.google.com finds that higher serum 25(OH)D concentration is associated with higher FEV1 among smokers and those with asthma, COPD, etc., but little effect on participants without respiratory diseases

Effect of MonthlyHigh-DoseLong-Term Vitamin D on Lung Function: A Randomized Controlled Trial.

Sluyter JD, Camargo CA, Waayer D, Lawes CMM, Toop L, Khaw KT, Scragg R.Nutrients. 2017 Dec 13;9(12):1353. doi: 10.3390/nu9121353.

The association of serum 25‐OH vitamin D with atopy, asthma, and lung function in a prospective study of anish adults

BH Thuesen, T Skaaby, LLN Husemoen… - Clinical & …, 2015 - Wiley Online Library

… 25(OH)D with decreased FEV1%pred but not with the FEV1/… the relation between vitamin
D levels and lung function in 14 … 25(OH)D with FEV 1 , but no association with the FEV 1 /FVC …

Save Cite Cited by 62

Serum vitamin D is associated with improved lung function markers but not with prevalence of asthma, emphysema, and chronic bronchitis

V Ganji, A Al-Obahi, S Yusuf, Z Dookhy, Z Shi - Scientific Reports, 2020 - nature.com

… (OH)D concentration is associated with improved lung function … if the vitamin D supplementation
improves lung function in … )D concentrations are directly associated with FCV and FEV1. …

Save Cite Cited by 11

Vitamin D deficiency, smoking, and lung function in the Normative Aging Study

NE Lange, Sparrow, P Vokonas… - … journal of respiratory …, 2012 - atsjournals.org

… smoke and vitamin D (20, 21), we investigated whether there was effect modification by VDD
on the relationship between smoking and lung … plotted FEV 1 simultaneously with vitamin D …

Save Cite Cited by 147

Air pollution in the urban location (Sao Paulo?) could account for some of the observed differences;

From Scholar.google.com

Health impact assessment of air pollution in São PauloBrazil

KC Abe, SGEK Miraglia - … journal of environmental research and public …, 2016 - mdpi.com

… Epidemiological research suggests that air pollution may cause chronic diseases, as well …
air pollution scenarios considering a Health Impact Assessment approach in São PauloBrazil. …

Save Cite Cited by 87 Related articles All 12 versions 

[PDF] od.lk

Air pollution and mortality in São PauloBrazil: Effects of multiple pollutants and analysis of susceptible populations

MA BravoJ Son, CU De Freitas, N Gouveia… - Journal of exposure …, 2016 - nature.com

… To our knowledge, the SES indicator used in this study has not been used in previous air
pollution epidemiology research in São Paulo, although earlier studies used other indicators. …

Save Cite Cited by 80 Related articles All 11 versions

[PDF] nih.gov

Air pollution and deaths among elderly residents of Sao PauloBrazil: an analysis of mortality displacement

AF Costa, G Hoek, B Brunekreef… - Environmental health …, 2017 - ehp.niehs.nih.gov

… of air pollution for periods of 1 month. Our findings suggest that in elderly residents of São
Paulo, nonaccidental mortality was displaced by < 30 days by the three pollutants studied. …

Save Cite Cited by 97 Related articles All 15 versions

[HTML] nih.gov

COVID-19 pandemic: Impacts on the air quality during the partial lockdown in São Paulo state, Brazil

LYK NakadaRC Urban - Science of the Total Environment, 2020 - Elsevier

… São Paulo state government. The aim of this study was to assess impacts on air quality in São
Paulo – Brazil… data from four air quality stations in São PauloBrazil to assess air pollutant …

Save Cite Cited by 575 Related articles All 14 versions

Incidence and mortality for respiratory cancer and traffic-related air pollution in São PauloBrazil

AG Ribeiro, GS Downward, CU de Freitas… - Environmental …, 2019 - Elsevier

… To quantify the association between exposures to traffic related air pollution and respiratory
cancer incidence and mortality within 
São PauloBrazil. Further, we aim to investigate the …

Save Cite Cited by 43 Related articles All 11 versions

[PDF] surrey.ac.uk

Air quality in the megacity of São Paulo: Evolution over the last 30 years and future perspectives

M de Fatima AndradeP KumarED de Freitas… - Atmospheric …, 2017 - Elsevier

… In 1975, CETESB was charged with monitoring air pollution and an air quality monitoring
network was established. The monitoring of 
SO 2 emissions by stationary sources began in …

Save Cite Cited by 188

Significant digits. The general rule is that no more non-zero digits should be given than are justified by the uncertainty of the value.

See "Too many digits: the presentation of numerical data"

https://www.ncbi.nlm.nih.gov/pmc/articles/PMC4483789/

If the uncertainty is greater than about 7%, only two non-zero digits are justified.

P values should be given to two decimal places unless the first two are 00 or the number lies between 0.045 and 0.054. If the first two are 00, then only one non-zero digit can be given.

Thus percentages for the 84 participants should be given in whole numbers

BMI (Kg/m2)

25.36 (±3.81)

Should be

BMI (Kg/m2)

25 (±4)

Please review all numbers in abstract, text, tables, and figures and adjust accordingly.

Author Response

Response to Reviewer 2 Comments

Revisor 2#

Dear reviewer, we greatly appreciate all the suggestions you have made regarding our manuscript. And we know that they all contributed to making the data better understood.

We have inserted the point-by-point responses into this letter. And we inserted convenient suggestions directly into the manuscript so that the article was satisfactory.

Point 1: I suggest making scatter plots of:

HGS vs. 25(OH)D

Muscle power vs. 25(OH)D

MIP and MEP vs. 25(OH)D

With urban and rural values indicated by different symbols

 Response 1: Scatter plots were made, as suggested, and the data were entered into the article and also submitted as supplementary material. (Figure 2).

Point 2: Please state the locations of the two populations in terms of the name of the city or rural region and latitude/longitude.

 Response 2: Below is the geographic location of the regions in which the study was carried out.

Latitude: -6.88634, Longitude: -38.5614

6° 53′ 11″ South, 38° 33′ 41″ West (urban area).

Latitude: -6.76782, Longitude: -38.2099

6° 46′ 4″ South, 38° 12′ 36″ West (rural area).

Point 3: A discussion of factors that affect handgrip strength for the elderly should be given.

Response 3: We have inserted a discussion paragraph with the reviewer's suggestions so that the strength, HGS, and PImax and Pemax data could be better understood. (Page 9, lines 266 - 276)

Round 2

Reviewer 1 Report

Except vitamin D supplementation (but in summer it should be without) all points are solved.

Author Response

Dear reviewer,

Thanks for your evaluation. Here is the answer to the question that you sent us.

Question #1: Except vitamin D supplementation (but in summer it should be without) all points are solved.

Point 1: In response to your statement, supplementation should ideally be given to individuals of all ages when vitamin D levels are below normal. However, our data suggest that in elderly people with muscle weakness, supplementation should be done continuously independent of seasonality.

Reviewer 2 Report

The scatter plots in Figure 3 are a good addition to the manuscript.

COBAS 8000 106 equipment (Roche® ).

Please give location of the company.

As I stated in my first review:

Significant digits. The general rule is that no more non-zero digits should be given than are justified by the uncertainty of the value.

See "Too many digits: the presentation of numerical data"

https://www.ncbi.nlm.nih.gov/pmc/articles/PMC4483789/

If the uncertainty is greater than about 7%, only two non-zero digits are justified.

P values should be given to two decimal places unless the first two are 00 or the number lies between 0.045 and 0.054. If the first two are 00, then only one non-zero digit can be given.

Thus, percentages should be in whole numbers

Age (years) (Mean±SD) 68.29 (±6.34) 70.53 (±8.38)

Should be

Age (years) (Mean±SD) 68 (±6) 71 (±8)

BMI (Kg/m2 ) 25.36 (±3.81)

Should be

BMI (Kg/m2 ) 25 (±4)

Please review all numbers in abstract, text, tables, and figures and adjust accordingly.

Author Response

Dear reviewer,

We have reviewed all of the outcome numbers in the manuscript and have categorically followed your suggestions.

I hope that the manuscript is as expected by you.
Our respects regards.